# Socio-Demographic, Lifestyle, and Clinical Characteristics of Early and Later Weight Status in Older Adults: Secondary Analysis of the ASPREE Trial and ALSOP Sub-Study

**DOI:** 10.3390/geriatrics8040071

**Published:** 2023-06-29

**Authors:** Tagrid A. Alharbi, Alice J. Owen, Joanne Ryan, Danijela Gasevic, John J. McNeil, Robyn L. Woods, Mark R. Nelson, Rosanne Freak-Poli

**Affiliations:** 1School of Public Health and Preventive Medicine, Monash University, 553 St. Kilda Rd, Melbourne, VIC 3004, Australia; 2Usher Institute, University of Edinburgh, Teviot Place, Edinburgh EH8 9AG, UK; 3Menzies Institute for Medical Research, University of Tasmania, 17 Liverpool St, Hobart, TAS 7001, Australia; 4School of Clinical Sciences at Monash Health, Monash University, 27-31 Wright Street, Melbourne, VIC 3004, Australia

**Keywords:** weight status, obesity, older adults, later life

## Abstract

Objective: To identify the socio-demographic, lifestyle, and clinical characteristics associated with self-reported weight status in early (age 18 years) and late (age ≥ 70 years) adulthood. Methods: The number of participants was 11,288, who were relatively healthy community-dwelling Australian adults aged ≥70 years (mean age 75.1 ± 4.2 years) in the Aspirin in Reducing Events in the Elderly (ASPREE) Longitudinal Study of Older Persons (ALSOP) sub-study. Self-reported weight at the study baseline (age ≥ 70 years) and recalled weight at age 18 years were collected. Height measured at baseline was used to calculate the BMI at both time points. Individuals were categorised into one of five ‘lifetime’ weight status groups: healthy weight (at both age 18 year and ≥70 years), overweight (at either or both times), non-obese (age 18 year) to obesity (age ≥70 years), obesity (age 18 years) to non-obese (age ≥ 70 years), and early and later life obesity (at age 18 years and ≥70 years). Results: Participants who experienced obesity in early and/or late adulthood were at a higher risk of adverse clinical characteristics. Obesity in late adulthood (regardless of early adulthood weight status) was associated with high proportions of hypertension, diabetes, and dyslipidaemia, whereas obesity in early adulthood (regardless of late adulthood weight status) was associated with lower cognitive scores (on all four measures). Discussion/Conclusion: Healthy or overweight weight status in early and later adulthood was associated with more favourable socioeconomic, lifestyle, and clinical measures. Obesity in early adulthood was associated with lower cognitive function in later adulthood, whereas obesity in later adulthood was associated with hypertension, diabetes, and dyslipidaemia.

## 1. Introduction

The global population is ageing, and the proportion of people aged ≥60 years has grown rapidly in recent decades [1]. The proportion of people aged ≥60 years is expected to increase from 1 billion in 2020 to 2.1 billion by 2050 [1]. This trend is partly driven by the increasing life expectancy, which does not always confer an improvement in healthy and disability-free life years among older people [2,3]. Moreover, the growth of the world’s ageing population means that healthcare systems are facing significant challenges in managing age-related chronic diseases. Therefore, understanding the risk factors for chronic, life-limiting diseases is important to promote health and well-being in older age and to prevent the burden on the health system.

The global prevalence of obesity has risen dramatically, irrespective of gender and age group [4]. Obesity is a modifiable risk factor for physical disability, major chronic diseases, and overall mortality and is increasing more rapidly in the older population [5,6]. As people are living longer, societies are faced with an increased burden of age-related chronic diseases, which are attributed to and exacerbated by obesity [5,6]. Adult weight status can fluctuate across the lifespan. Body weight tends to rise during early and middle adulthood [7]; however, an understanding of the characteristics associated with body weight is limited by its collection at a single time point in epidemiological studies [8]. Excess weight across the lifespan can lead to a variety of negative health risks via a variety of mechanisms and pathways. Excessive adipose tissue can lead to increased insulin resistance [9], elevated free fatty acids [9], and systemic inflammation [10], all of which contribute to the development and progression of cardiometabolic disorders, such as type 2 diabetes, hypertension, and cardiovascular disease (CVD) [9,10]. Along with cardiometabolic disorders, which may involve the same metabolic pathways as adiposity, obesity has also been associated with an increased risk of non-metabolic diseases, such as cognitive impairment and dementia [11]. These adverse health consequences may lead to impairments in physical function [12] and disability [13], lower quality of life [14], and an increased risk of mortality [14,15].

A recent study that examined body weight status in early (age 18 years) and later (age ≥ 70 years) adulthood revealed that the risk of mortality doubled for older adults who were obese in both early and later adulthood, compared to older adults who had a healthy body mass index (BMI) in both early and later adulthood [16]. These findings highlight the importance of weight management interventions in public health. For such interventions to be effective, they need to be relevant and accessible to those who are targeted [17]. To develop effective weight management interventions, a better understanding of the socio-demographic and clinical characteristics of overweight/obese target population groups is needed.

This study aims to identify the socio-demographic and lifestyle characteristics associated with self-reported weight status in early (age 18 years) and late (age ≥ 70 years) adulthood.

## 2. Materials and Methods

This is a secondary analysis utilising data from the Aspirin in Reducing Events in the Elderly (ASPREE) clinical trial and the ASPREE Longitudinal Study of Older Persons (ALSOP) sub-study, which have been described previously [9,10,11,12].

### 2.1. Study Population

ASPREE was a double-blind, randomised controlled trial of low-dose aspirin on disability-free survival in 19,114 healthy older adults living independently in the United States (age ≥ 65 years) and Australia (age ≥ 70 years). At study entry, the participants were free of known cardiovascular diseases, dementia, or life-limiting serious conditions. The majority (89%) of Australian ASPREE participants (*n* = 14,892) participated in the ALSOP sub-study. A small number (*n* = 513, 3.4%) did not self-report their weight at baseline, and the weight measured at ASPREE baseline was used in place of the missing self-reported weight (*n* = 474). Participants who were categorised as underweight at age ≥ 70 years (*n* = 110) were excluded, as underweight status in older age is considered a clinical risk factor for mortality [8,18]. In total, we obtained complete weight and height data on 11,288 participants (mean age 75.1 ± 4.2 years, 54.5 % women), representing our sample (Appendix A).

### 2.2. Weight Status

Self-reported body weight in early (18 years) and late (study entry, ≥70 years) adulthood was converted to BMI (kg/m^2^) using height measured at ASPREE enrolment [19]. BMI categories were defined according to the World Health Organization (WHO) cut-offs: healthy weight (≥18.5–24.9 kg/m^2^), overweight (25.0–29.9 kg/m^2^), and obesity (≥30 kg/m^2^). The ALSOP cohort was >90% Caucasian [20]; therefore, these cut points are justified.

Lifetime weight status was defined as follows: healthy weight (both age 18 and ≥70 years), overweight (either or both times), non-obese (age 18 years) to obesity (age ≥ 70 years), obesity (age 18 years) to non-obese (age ≥ 70 years), and early and later life obesity (age 18 years and ≥70 year).

### 2.3. Potential Associated Characteristics

Characteristics were selected a priori based on previously identified factors associated with weight status [21] (Appendix A). In summary, we included socio-demographic (age, gender, living situation, partner status, education, rurality, residential suburb socioeconomic status, and work), lifestyle behaviours (physical activity, smoking, alcohol, and social health), and health (depressive symptoms, health-related quality of life, hypertension, diabetes, dyslipidaemia, and cognitive function) characteristics.

### 2.4. Statistical Analysis

Potential associated characteristics were examined by self-reported weight status categories using the Chi-square test (categorical) and ANOVA or Wilcoxon rank-sum tests (continuous).Interaction effects of gender were assessed, as women have a higher body fat percentage and different body fat distribution than men [22]. All statistical analyses were conducted using thesoftware STATA Version 15.1 (StataCorp LLC., College Station, TX, USA).

## 3. Results

Healthy weight (*n* = 3494, 30.9%) was associated with the most beneficial socio-demographic, health behaviours, and clinical outcomes, as shown in Table 1 and Appendix A. Participants in the overweight category (*n* = 5135, 45.5%) were similar to participants in the healthy weight category concerning the health behaviours and clinical outcomes assessed. However, the demographic profile of participants in the overweight category was different, with the highest proportion of participants who were men, partnered, and living at home with family/friends.

The three categories of participants who experienced obesity in early and/or late adulthood had a greater burden of clinical risk factors. Obesity in late adulthood (regardless of early adulthood weight status) was associated with a lower physical health-related quality of life, and high proportions of hypertension, diabetes, and dyslipidaemia. Obesity in early adulthood (regardless of late adulthood weight status) was associated with lower cognitive scores (on all four measures).

The two lifetime weight categories were identified as having a high mortality risk in our previous study: (1) participants who were in the early and later life obesity category were associated with increased mortality risk, and (2) the risk was even greater for participants who were in the obesity to non-obesity category [16]. Compared to other weight status categories, participants who were in the early and later life obesity category (1.2%) were more likely to be younger, unpartnered, live in an outer/remote region, have lower residential suburb socioeconomic status (SEIFA), be in paid work, not engage in volunteer work, do less physical activity in middle age and later life, be a current or former smoker, have a lower physical health-related quality of life, and more likely to have hypertension, diabetes, dyslipidaemia, and lower cognitive scores (on all four measures). Participants who were in the obesity to non-obesity category (0.8%) were more likely to be older, live alone, be unpartnered, have lower education, live in a major city, do volunteer work, do less physical activity in middle age but more in later life, have never smoked, never drank alcohol, and have high depressive symptoms and lower cognitive scores (on all four measures).

Four gender interactions were identified (Table 1 and Table 2). Within the non-obesity to obesity category, women were more likely to live in socioeconomically advantaged suburbs and men were more likely to live in least advantaged suburbs compared to other weight status categories. Men generally undertook more physical activity in later life than women, and this was most evident in the three obesity categories. Men tended to drink alcohol at high-risk levels, whereas women had a greater proportion of never drinking alcohol. These alcohol differences were most evident in the early and later life obesity category (never drinkers: 6.1% men, 35.8% women; high-risk: 39.4% men, 4.5% women). Men tended to self-report higher physical health-related quality of life compared to women, except within the early and later life obesity category. For cognitive performance, the only gender interaction was for the COWAT measure, where men tended to have lower scores, especially within the three obesity categories, compared to women.

## 4. Discussion

In this large population-based study of community-dwelling older adults aged ≥70 years in good health, we observed that obesity in early adulthood was associated with lower cognitive scores, whereas obesity in late adulthood was associated with lower physical health-related quality of life, hypertension, diabetes, and dyslipidaemia. Older adults who were either healthy weight or overweight in early and late adulthood had better health outcomes later in life. Of all the demographic characteristics explored, only residential suburban socioeconomic disadvantage was associated with lifetime obesity or non-obesity to obesity group.

We observed that early obesity (regardless of obesity status in later life) was associated with lower cognitive function across four domains, which aligns with the findings of neuropsychological studies suggesting that early adult and middle-aged obese individuals performed lower on higher-order cognitive function tasks than healthy weight individuals [23,24]. Similarly, another study observed that a stable BMI over time was associated with a reduced risk of poorer cognitive outcomes [25]. As obesity is a modifiable risk factor for cognitive decline, early identification and management could help minimise the risk of poor cognition in later life.

There was an association between obesity in later adulthood (regardless of obesity status in early life) and the chronic conditions that increase cardiovascular disease risk of hypertension, diabetes, and dyslipidaemia. These findings are unsurprising, as obesity is known to increase the risk of ill health, including these conditions. Previous research has also shown a dose–response relationship between years of living with obesity and increased risk of diabetes [26,27]. However, in our study of relatively healthy older adults, the proportions of participants with hypertension, diabetes, and dyslipidaemia were similar to those who were obese in later life, regardless of their obesity status at 18 years of age. Therefore, obesity continues to have negative effects later in life. The present study provides evidence that, among adults aged ≥70 years in reasonably good health, being overweight has similar health outcomes to being a healthy weight. Similarly, our previous analysis in this cohort found that overweight and healthy-weight participants had a similar risk of all-cause mortality [16]. In fact, being overweight in late adulthood may be protective, as meta-analyses on the BMI–mortality relationship among older adults reported reduced risks of all-cause mortality for overweight status participants compared to those with a healthy weight [6,28]. Hence, the BMI–mortality relationship is often described as U-shaped. Our findings add to the evidence that the WHO BMI classification for body weight status may need to be redefined for older adults to include overweight as part of healthy weight [29]. The findings of our study, along with those of other studies, suggest that being overweight may increase longevity in older age [8,30], though further research is needed to determine the extent of the benefit.

While the majority of associations were similar across binary gender, there were a few differences. Specifically, the most socioeconomically advantaged women were more likely to be in the non-obesity to obesity category, whereas it was the opposite (the least advantaged) for men. Generally, we observed that men were more likely to undertake physical activity in later life, consume alcohol at levels defined as high-risk, self-report higher physical health-related quality of life, and have lower cognitive scores than women. In each case, these gender differences were most evident in the early and/or later life obesity categories and least evident in the healthy body weight category. Hence, there was a gender–body weight–health interaction between these lifestyle characteristics and cognitive functioning. These gender differences are likely indicative of the broader socio-cultural and biological differences between men and women, which are becoming increasingly apparent [22,31].

### Strengths and Limitations

To the best of our knowledge, our study is the first to explore the associations between socio-demographic and lifestyle characteristics and changes in self-reported weight status in early and later life. Our results are broadly generalisable to people who reach the age of 70+ years, live independently, and are free of dementia, major physical disability, or life-limiting disease within five years. Survival bias, which occurs when individuals with a condition (i.e., obesity) are less likely to participate in a study (due to death or being severely ill) compared to those who do not have the condition (i.e., healthy weight), needs to be considered when interpreting our findings. There is other evidence that lifetime obesity is associated with increased health issues, which can (1) lead to premature death before our baseline recruitment at age ≥ 70 years, and (2) increase the risk of cardiovascular disease, dementia, or a life-limiting disease that would preclude their eligibility for enrollment into the ASPREE primary prevention clinical trial. Therefore, the health consequences of obesity may have indirectly selected participants in our sample, who may differ from those with obesity in the community in terms of obesity severity and its impact on health-related issues. For example, our sample only contains 2% of participants with early-life obesity. Our findings also suggest that those who were obese in early life, but not in later life, have adapted their health-related behaviours. We observed that participants who were no longer obese were more likely to engage in physical activity in later life, have never smoked, and have never taken alcohol. Hence, participants aged ≥70 years with obesity in early life without overt diseases likely represent a healthier cohort than the general population.

The main limitation of this study is its cross-sectional design, limiting the ability to determine causality between the associated characteristics and weight status in early and later life. Another limitation is that body weight was self-reported, and body weight at age 18 was recalled at age ≥ 70. However, a recent meta-analysis demonstrated that the recall of early life weight might be a valid measure to use in the epidemiological analysis [18]. Another limitation is that the retrospective BMI at age 18 was calculated using objectively measured height at age 70. Most people reach their adult height before the age of 18, but they usually lose some height as they age. As a higher height would produce a lower BMI category, some participants may be miscategorised as obese instead of overweight, or overweight instead of healthy weight. Therefore, our findings for the obese categories are more likely to be conservative. Finally, the measurement of dietary intake was not available for this cohort. Among all age cohorts, dietary intake is strongly associated with weight status [32,33,34,35]. In this study, we hypothesise that dietary intake could have gender–body weight–health interactions, similar to our physical activity findings.

## 5. Conclusions

Our findings demonstrate the health outcomes of lifetime weight status for people who reach the age of 70 and are in relatively good health, which can inform the development of interventions for maintaining a healthy weight and preventing obesity over an adult lifespan. Community-dwelling adults aged ≥70 years with a healthy weight or being overweight are more likely to have favourable socio-demographic conditions, health-related behaviours, and clinical measures. Being obese early or later in life carries a greater health burden, with obesity in both early and later life carrying the greatest health burden. Our findings are likely conservative, given that the survival bias and the healthy volunteer effect limited the most severely obese or ill from being included in the cohort. While the majority of associations were similar across genders, we observed that gender–body weight–health interactions are indicative of broader socio-cultural and biological differences between men and women.

This study highlights that the lifetime weight status can contribute to health risks. Obesity prevention or treatment throughout life will have health benefits in later life. Maintaining a healthy weight throughout adulthood and/or being overweight in later life will also have health benefits in later life. We highlight that only one demographic characteristic, the suburban socioeconomic disadvantage, was associated with lifetime obesity or non-obesity to obesity groups. Hence, people living in lower socioeconomic suburbs should be targeted to maintain a healthy weight and prevent obesity over an adult lifespan.

## Figures and Tables

**Table 1 geriatrics-08-00071-t001:** Characteristics of participants by early and later life weight status (*n* = 11,288).

Early (18 years) and Later Life (≥70 years) Weight Status * (Mean ± SD or n (%) or Median (IQR))							
Characteristic	Healthy Weight	Overweight	Non-Obesity to Obesity	Obesity to Non-Obesity	Early and Later Life Obesity	*p*-Value	*p*-Value Gender Interaction
N	3494 (30.9%)	5135 (45.5%)	2431 (21.6%)	95 (0.8%)	133 (1.2%)		
**Demographic Factors**							
Age years mean ± SD	75.5 ± 4.5	75.1 ± 4.3	74.5 ±3.7	76.6 ± 5.1	74.6 ± 3.7	<0.001	0.14
**Gender**							
Men	1350 (38.6%)	2725 (53.1%)	957 (39.4%)	42 (44.2%)	66 (49.6%)	<0.001	0.41
Women	2144 (61.4%)	2410 (46.9%)	1474 (60.6%)	53 (55.8%)	67 (50.4%)		
**Living situation**							
At home with family/friends	2359 (67.5%)	3724 (72.5%)	1658 (68.2%)	51 (53.7%)	91 (68.4%)	<0.001	0.42
At home alone	1135 (32.5%)	1411 (27.5%)	773 (31.8%)	44 (46.3%)	42 (31.6%)		
**Partner status**							
Partnered	1943 (64.2%)	3138 (70.1%)	1300 (63.1%)	45 (54.9%)	66 (58.4%)	<0.001	0.22
Unpartnered	1084 (35.8%)	1338 (29.9%)	762 (36.9%)	37 (45.1%)	47 (41.6%)		
**Years of education**							
<12 years	1993 (57.0%)	3072 (59.8%)	1595 (65.6%)	65 (68.4%)	89 (66.9%)	<0.001	0.80
>12 years	1501 (43.9%)	2063 (40.2%)	835 (34.3%)	30 (31.5%)	44 (33.1%)		
**Region**							
Major city	1204 (34.6%)	1869 (36.5%)	890 (36.7%)	41 (43.1%)	44 (33.1%)	<0.001	0.19
Inner region	1912 (54.9%)	2629 (51.4%)	1224 (50.5%)	45 (47.4%)	63 (47.4%)		
Outer/remote regional	367 (10.5%)	620 (12.1%)	309 (12.8%)	9 (9.5%)	26 (19.5%)		
**Economic factors**							
**Area level socioeconomic status (SEIFA)**							
Least advantaged	1050 (30.2%)	1694 (33.1%)	919 (37.9%)	32 (33.7%)	58 (43.6%)	<0.001	0.005
Middle tertial	1289 (37.0%)	1971 (38.5%)	910 (37.6%)	44 (46.3%)	49 (36.8%)		
Most advantaged	1144 (32.8%)	1453 (28.4%)	594 (24.5%)	19 (20.0%)	26 (19.6%)		
**Paid work**							
No	2722 (91.9%)	3948 (89.9%)	1850 (91.2%)	71 (88.8%)	89 (85.6%)	0.013	0.39
Full-time/part-time	240 (8.1%)	444 (10.1%)	179 (8.8%)	9(11.3%)	15 (14.4%)		
**Volunteer work**							
No	1604 (53.5%)	2586 (58.6%)	1231 (60.3%)	38 (47.5%)	70 (63.1%)	<0.001	0.09
Yes	1397 (46.5%)	1828 (41.4%)	810 (39.7%)	42 (52.5%)	41 (36.9%)		
**Health-Related Behaviours**							
**Physical activity in middle age**							
Never/rarely/light	279 (9.2%)	450 (10.1%)	231 (11.3%)	12 (14.8%)	16 (14.8%)	0.035	0.015
Moderate/vigorous	2748 (90.8%)	4010 (78.1%)	1815 (88.7%)	69 (85.2%)	92 (85.2%)		
**Physical activity at ≥70 years**							
Never/rarely/light	783 (25.9%)	1325 (29.8%)	981 (47.9%)	23 (29.1%)	53 (49.1%)	<0.001	<0.001
Moderate/vigorous	2241 (74.1%)	3120 (70.2%)	1063 (52.1%)	56 (70.9%)	55 (50.9%)		
**Smoking status**							
Never	2090 (59.8%)	2727 (53.1%)	1271 (52.3%)	57 (60.0%)	65 (48.9%)	<0.001	0.07
Former/Current	1404 (40.2%)	2408 (46.9%)	1160 (47.7%)	38 (40.0%)	68 (51.1%)		
**Alcohol consumption**							
Never	514 (14.7%)	709 (13.8%)	432 (17.8%)	23 (24.2%)	28 (21.1%)	<0.001	<0.001
Former/Current—Low Risk	2067 (59.2%)	2993 (58.3%)	1490 (61.3%)	54 (56.8%)	76 (57.1%)		
Current—High Risk	913 (26.1%)	1433 (27.9%)	509 (20.9%)	18 (19.0%)	29 (21.8%)		
**Social health**							
Positive	3257 (93.2%)	4794 (93.4%)	2217 (91.2%)	87 (91.6%)	124 (93.2%)	0.011	0.46
Lonely/socially isolated/and/or low social support	237 (6.8%)	340 (6.6%)	214 (8.8%)	8 (8.4%)	9 (6.8%)		
**Depressive symptoms**							
<8 low depressive symptoms	3211 (91.9%)	4703 (91.6%)	2140 (88.1%)	82 (86.3%)	118 (88.7%)	<0.001	0.53
≥8 high depressive symptoms	283 (8.1%)	432 (8.4%)	291 (11.9%)	13 (13.7%)	15 (11.3%)		
**Health-related Quality of Life** median (IQR)							
Mental component score	57.2 (52.3–60.0)	57.2 (52.9–60.2)	57.3 (52.3–61.4)	57.2 (52.4–60.9)	57.1 (52.3–61.1)	0.0086	0.60
Physical component	52.5 (46.2–56.1)	51.0 (44.0–55.5)	46.0 (38.3–52.5)	49.6 (43.2–55.1)	43.1 (38.1–51.1)	<0.001	<0.001
**Clinical Measures**							
**Hypertension**							
No	1221 (34.9%)	1281 (24.9%)	372 (15.3%)	24 (25.3%)	22 (16.5%)	<0.001	0.85
Yes	2273 (65.1%)	3854 (75.1%)	2059 (84.7%)	71 (74.7%)	111 (83.5%)		
**Diabetes mellitus**							
No	3328 (95.3%)	4686 (91.3%)	2016 (82.9%)	87 (91.6%)	104 (78.2%)	<0.001	0.22
Yes	166 (4.7%)	449 (8.7%)	415 (17.1%)	8 (8.4%)	29 (21.8%)		
**Dyslipidaemia**							
No	1153 (33.0%)	1640 (31.9%)	717 (29.5%)	33 (34.7%)	38 (28.6%)	0.054	0.67
Yes	2341 (67.0%)	3495 (68.1%)	1714 (70.5%)	62 (65.3%)	95 (71.4%)		
**Cognitive performance**							
3MS	93.9 ± 4.42	93.6 ± 4.38	93.6 ± 4.36	92.5 ± 5.20	93.4 ± 4.34	0.0013	0.63
COWAT	12.7 ± 4.61	12.2 ± 4.65	11.9 ± 4.53	10.9 ± 4.23	11.4 ± 4.31	<0.001	0.048
SDMT	37.5 ± 10.03	37.2 ± 9.93	36.9 ± 9.59	33.6 ± 10.47	34.3 ± 8.74	<0.001	0.58
HVLT-R Delayed Recall	5.7 ± 2.07	5.6 ± 1.98	5.7 ± 1.98	5.2 ± 1.97	5.4 ± 2.02	0.006	0.77

* Early to later life weight status excludes individuals underweight at age 70. Healthy weight (at both age 18 and ≥70 years), overweight (at either or both times), non-obese (age 18 years) to obesity (age ≥ 70 years), obesity (age 18 years) to non-obese (age ≥ 70 years), and early and later life obesity (at age 18 and ≥70 years). *n* = number of observations; SD = standard deviation; *p*-Values are from ANOVA or Wilcoxon rank-sum tests (continuous), or Chi-square tests (categorical variables). SEIFA: the socioeconomic indexes for areas based on the Index of Relative Socioeconomic Advantage and Disadvantage, which is based on residential postcode. Depressive symptoms: Center for Epidemiological Studies–Depression. Health-related Quality of Life (SF-12). Hypertension: high blood pressure measurement of systolic ≥ 140 mmHg or diastolic ≥ 90 mmHg or anti-hypertensive medications use. Diabetes: self-report of diabetes or fasting glucose ≥ 126 mg/dL or diabetes medications. Dyslipidaemia: cholesterol-lowering medications or high serum cholesterol (total ≥ 212 mg/dL, or high-density lipoprotein ≥ 240 mg/dL or low-density lipoprotein >160 mg/dL). 3MS: Modified Mini-Mental State Examination. COWAT: Controlled Oral Word Association Test. SDMT: Symbol Digit Modalities Test. HVLT-R: the delayed recall score: Hopkins Verbal Learning Test—Revised.

**Table 2 geriatrics-08-00071-t002:** Gender disaggregated findings: Characteristics of participants by early and later life weight status (*n* = 11,288).

Characteristic	Healthy Weight	Overweight	Non-Obesity to Obesity	Obesity to Non-Obesity	Early and Later Life Obesity	*p*-Value
**Women**						
**SEIFA**						
Least advantaged	642 (30.0%)	803 (33.5%)	563 (38.3%)	20 (37.7%)	34 (50.7%)	<0.001
Middle tertial	788 (36.8%)	936 (39.0%)	552 (37.5%)	26 (49.1%)	27 (40.4%)	
Most advantaged	708 (33.1%)	661 (27.5%)	356 (34.2%)	7 (13.2%)	6 (8.9%)	
**Physical activity at ≥70 years**						
Never/rarely/light	534 (28.9%)	793 (38.3%)	675 (54.4%)	15 (37.5%)	30 (57.7%)	<0.001
Moderate/vigorous	1314 (71.1%)	1279 (61.7%)	566 (45.6%)	25 (62.5%)	22 (42.3%)	
**Alcohol consumption**						
Never	399 (18.6%)	486 (20.2%)	364 (24.7%)	19 (35.8%)	24 (35.8%)	<0.001
Former/Current—Low Risk	1309 (61.1%)	1516 (62.9%)	917 (62.2%)	27 (50.9%)	40 (59.7%)	
Current—High Risk	436 (20.3%)	408 (16.9%)	193 (13.1%)	7 (13.2%)	3 (4.5%)	
SF-12 Physical component, median (IQR)	52.1 (45.4–56.1)	49.3 (42.4–54.8)	44.4 (37.0–51.2)	49.7 (43.2.3–54.7)	41.4 (36.0–51.4)	<0.001
COWAT Overall Score, mean ± SD	13.2 ± 4.58	12.9 ± 4.64	12.3 ± 4.40	10.8 ± 4.25	11.9 ± 4.15	<0.001
**Men**						
**SEIFA,** n (%)						
Least advantaged	408 (30.3%)	891 (32.8%)	356 (37.4%)	12 (28.6%)	24 (36.4%)	0.009
Middle quintile	501 (37.2%)	1035 (38.1%)	358 (37.6%)	18 (42.8%)	22 (33.3%)	
Most advantaged	436 (32.4%)	792 (29.1%)	238 (25.0%)	12 (28.6%)	20 (30.3%)	
**Physical activity at ≥70 years**						
Never/rarely/light	249 (21.2%)	532 (22.4%)	306 (38.1%)	8 (20.5%)	23 (41.1%)	<0.001
Moderate/vigorous	927 (78.8%)	1841 (77.6%)	497 (61.9%)	31 (79.5%)	33 (58.9%)	
**Alcohol consumption**						
Never	115 (8.5%)	223 (8.2%)	68 (7.1%)	4 (9.5%)	4 (6.1%)	0.13
Former/Current—Low Risk	758 (56.2%)	1477 (54.2%)	573 (59.9%)	27 (64.3%)	36 (54.5%)	
Current—High Risk	477 (35.3%)	1025 (37.6%)	316 (33.0%)	11 (26.2%)	26 (39.4%)	
SF-12 Physical component, median (IQR)	52.9 (47.2–56.1)	52.2 (45.6–56.1)	48.5 (40.7–53.9)	49.6 (43.1–55.1)	44.2 (40.1–51.0)	<0.001
COWAT Overall Score, mean ± SD	12.0 ± 4.56	11.6 ± 4.56	11.2 ± 4.66	11.0 ± 4.26	11.0 ± 4.45	0.001

Healthy weight (at both age 18 and ≥70 years), overweight (at either or both times), non-obese (age 18 years) to obesity (age ≥ 70 years), obesity (age 18 years) to non-obese (age ≥ 70 years), and early and later life obesity (at age 18 and ≥70 years). *n* = number of observations; SD = standard deviation; *p*-Values are from ANOVA or Wilcoxon rank-sum tests (continuous), or Chi-square tests (categorical variables). SEIFA: the socioeconomic indexes for areas based on the Index of Relative Socioeconomic Advantage and Disadvantage, which is based on residential postcode. COWAT: Controlled Oral Word Association Test, SF-12: Health-related Quality of Life.

## Data Availability

The original contributions presented in the study are included in the article/Appendix A; further inquiries can be directed to the corresponding author/s.

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
