# Peer review of "Socio-Demographic, Lifestyle, and Clinical Characteristics of Early and Later Weight Status in Older Adults: Secondary Analysis of the ASPREE Trial and ALSOP Sub-Study"

_geriatrics, 2023, doi:10.3390/geriatrics8040071_

Round 1

Reviewer 1 Report

Dear Authors,

Thank you for your manuscript. The paper is well-written and presents interesting findings on factors associated with weight status at the age of 18 and 70 years from a large sample. Please see my comments below.

The Introduction should incorporate an additional paragraph explaining the mechanisms associated with the weight increase during the lifespan.

Also, I wonder why the authors do not provide multivariable-adjusted analyses and ORs, for example, a model of factors associated with obesity at age 70 with an initial healthy weight at age 18.

Next, dietary habits were not taken into account. It should be mentioned as an important limitation.

Minor comments. Please provide the mean age (±SD) of the study participants in the abstract and section "Study Population". According to MDPI formats, the subsections should be numbered as well.

Please revise the sentence in the Abstract, lines 29-30: ...was associated with hypertension, etc?

Overall, my comments do not detract from the work done and I think the article is worthy of publication.

Reviewer 2 Report

Thank you for the opportunity to review this manuscript. The objectives are clear and the research is of significance to public health. The sex-specific analyses are especially helpful. There are some minor issues with presentation, mostly from the point of view of readers that work within the sphere of older adults health but not specifically large data sets. This can have a wider readership with a few tweaks.

Title and Abstract: Please make it clearer that this cohort is made up of participants from two studies. Socio-demographic, lifestyle......: secondary analysis of the ASPREE and ALSOP study. 

Materials and methods: It can be confusing that the majority of the Australian ASPREE study also were participants of the ALSOP. May I suggest the authors to clarify this so that readers that are not familiar with the primary trials, also would be able to understand. While the study protocol and main results have been previously published, it will be helpful if the authors add in the total sample size.

Results:

Line 114-119 - please rephrase this to improve readability

Discussion and conclusion:

Well-discussed and the conclusions are not overstated. I have no issues with it.

Others:

1. Please also do a spellcheck e.g. line 91 "socio-economic"; line 92 "behaviours". Also punctuations, line 103 "n=5135, 45.5%)

2. Suggestion to change 'gender' to 'sex'; males and females, but not crucial
